# A Low-Carbon and Economic Dispatch Strategy for a Multi-Microgrid Based on a Meteorological Classification to Handle the Uncertainty of Wind Power

**DOI:** 10.3390/s23115350

**Published:** 2023-06-05

**Authors:** Yang Liu, Xueling Li, Yamei Liu

**Affiliations:** 1College of Electrical Engineering, Sichuan University, Chengdu 610065, China; yang.liu@scu.edu.cn (Y.L.); shirley_neenee@163.com (X.L.); 2Key Laboratory of Intelligent Electric Power Grid of Sichuan Province, Sichuan University, Chengdu 610065, China

**Keywords:** adjustable robust optimization, uncertainty, meteorological classification, low-carbon operation, economic operation

## Abstract

In a modern power system, reducing carbon emissions has become a significant goal in mitigating the impact of global warming. Therefore, renewable energy sources, particularly wind-power generation, have been extensively implemented in the system. Despite the advantages of wind power, its uncertainty and randomness lead to critical security, stability, and economic issues in the power system. Recently, multi-microgrid systems (MMGSs) have been considered as a suitable wind-power deployment candidate. Although wind power can be efficiently utilized by MMGSs, uncertainty and randomness still have a significant impact on the dispatching and operation of the system. Therefore, to address the wind power uncertainty issue and achieve an optimal dispatching strategy for MMGSs, this paper presents an adjustable robust optimization (ARO) model based on meteorological clustering. Firstly, the maximum relevance minimum redundancy (MRMR) method and the CURE clustering algorithm are employed for meteorological classification in order to better identify wind patterns. Secondly, a conditional generative adversarial network (CGAN) is adopted to enrich the wind-power datasets with different meteorological patterns, resulting in the construction of ambiguity sets. Thirdly, the uncertainty sets that are finally employed by the ARO framework to establish a two-stage cooperative dispatching model for MMGS can be derived from the ambiguity sets. Additionally, stepped carbon trading is introduced to control the carbon emissions of MMGSs. Finally, the alternative direction method of multipliers (ADMM) and the column and constraint generation (C&CG) algorithm are adopted to achieve a decentralized solution for the dispatching model of MMGSs. Case studies indicate that the presented model has a great performance in improving the wind-power description accuracy, increasing cost efficiency, and reducing system carbon emissions. However, the case studies also report that the approach consumes a relative long running time. Therefore, in future research, the solution algorithm will be further improved for the purpose of raising the efficiency of the solution.

## 1. Introduction

With the issues of global pollution and fossil-energy depletion becoming more and more severe, carbon-emission reduction has become an important focus for technological development [1]. As a result of its remarkable advantages [2], renewable energy generation has been adopted for low-carbon applications in the power system [3]. Research in this area [4] has provided electricity generated by renewable energy to the green cottage and obtained relatively greater benefits. However, the strong uncertainty and randomness of renewable energy generation cause significant issues in terms of the security and efficiency of the power system [5]. Microgrids are able to provide effective solutions for the local consumption of renewable energy owing to their flexible operation and high autonomy [6], avoiding the issues caused by the direct connections of renewable energies to the grid. Additionally, with the advancement of electricity marketization, it is possible to interconnect a certain number of microgrids to form an MMGS [7]. An MMGS aims to achieve a balance between power supply and demand by means of energy interaction and collaborative optimization within the system, further promoting the consumption of renewable energies. Compared to individual microgrids, MMGSs have better overall economic efficiency and lower carbon emissions [8]. However, the uncertainty of the energy generation in the microgrid, especially that produced by wind energy, may lead to poor energy utilization [9]. This leads to dispatching operation being dependent on units with a high carbon emission and a failure to fully utilize the low-carbon characteristics of the microgrid [10]. Therefore, it is highly valuable to study the dispatching strategy for MMGSs with regard to wind-power uncertainty to ensure carbon-emission reduction and economic improvement.

As mentioned above, the uncertainty of wind-power generation in the microgrid leads to difficulty in dispatching decisions. Mathematical approaches such as stochastic optimization (SO) and robust optimization (RO) are widely employed to address dispatching decisions involving uncertainties [11]. Li et al. [10] presented a scenario-based optimal operational model based on SO, mainly to handle the uncertainties involved in energy demand and renewable generation. The effectiveness of the approach was verified according to the experimental results. However, Chen et al. [12] pointed out that the probability distribution of the random variables in SO is difficult to obtain precisely. They also suggested that in the RO model, the range of renewable energy generation can be depicted by constructing an uncertainty set without information on the probability distribution of variables. Although RO is able to provide better dispatching strategies theoretically, Wang et al. [13] suggested that the decision results based on RO are frequently overly conservative. Therefore, Zhai et al. [14] presented an ARO model to address renewable energy uncertainty while utilizing the controllable uncertainty set to reduce the conservatism of the solution. Case studies illustrated that the ARO model can better balance the economy and robustness of the dispatching strategy by controlling the frequency of worst-case scenarios. However, the ignorance of the uncertainty probability information in ARO and the subjective determinations of the current construction of the uncertainty set result in a lack of precise delineation of the variables [15].

In order to address the shortage of ARO uncertainty intervals, historical data on wind power are employed to provide scientific data support for the construction of the uncertainty set [16]. Wu et al. [11] developed an ambiguity set of the probability distribution of wind power based on the imprecise Dirichlet model, which used the historical data to establish the uncertainty intervals without relying on subjective determinations. The model was adopted into the ARO framework, which objectively improved the efficiency of the final dispatching model. Wang et al. [17] constructed the ambiguity set of the probability distribution of wind power based on the Wasserstein distance, and the experimental results indicated that the description accuracy of the uncertainty gradually increased with the increasing number of historical samples. The research suggests that the construction of the ambiguity set highly depends on a sufficient amount of historical data.

Therefore, it is fundamental to generate a suitable amount of historical wind-power data for ARO to build a precise ambiguity set in terms of achieving the optimal dispatching strategy for microgrids. Jiang et al. [18] presented a generative adversarial network (GAN) to generate synthesized wind-power scenarios. Ning et al. [19] generated a set number of wind-power data samples using GAN to implement economic dispatching optimization. The experimental results of these studies demonstrate that GAN can accurately capture the probability-distribution characteristics of the sample data and generate new synthesized samples with similar statistical characteristics to expand the historical dataset. Meanwhile, Li et al. [20] and Qin et al. [21] reported that wind-power output is highly related to the meteorology in the location of the microgrids; thus, the distribution of wind power varies under different meteorological categories. Nevertheless, the existing research, which integrates data-generation techniques and the ambiguity set to reflect the uncertainty of wind-power generation, tends to assume that the distributions in the ambiguity set are similar. In addition, most studies employ only one uncertainty set to describe the range of variation. It should be pointed out that a single uncertainty set is not conducive to accurately portraying the uncertainty. Therefore, Xu et al. [8] clustered the historical meteorological dataset to obtain various meteorological patterns, thereby building a wind-power prediction model for different meteorological patterns. The simulations proved that the clustering model can adequately extract historical data to improve the prediction accuracy, with better results than the non-clustering model. Therefore, meteorological classification combined with the generation of data using GAN is an effective way to improve the utilization of data samples and the effectiveness of ARO in terms of achieving the optimal dispatching strategy.

Additionally, to further reduce carbon emissions, studies involving coordinated dispatching for MMGS in combination with carbon trading have been conducted. Zhang et al. [22] and Wang et al. [23] introduced the carbon-trading mechanism to MMGSs and constructed a decentralized dispatching model using ADMM. The simulation results demonstrated the effectiveness of the carbon-trading mechanism in guiding an MMGS to reduce carbon emissions. Yang et al. [24] achieved efficient low-carbon operation by introducing stepped carbon trading into an optimal dispatching mechanism, and the analysis showed a stronger carbon-reduction ability compared to the traditional carbon-trading approaches. Therefore, the further use of stepped carbon prices to limit carbon emissions as part of the dispatching strategy can additionally improve the low-carbon performance of MMGSs.

Motivated by previous research, this paper focuses on the efficient low-carbon dispatching of MMGSs in light of wind-power uncertainty using stepped carbon trading. The main contributions of this paper are summarized as follows: (i) According to the variability of wind power in different meteorological scenarios, the meteorological features that are highly relevant to wind-power output are determined. Then, a meteorological clustering model is established to improve the effectiveness of utilizing the historical wind-power data; (ii) a GAN-based synthesized data-generation technique and an ambiguity set are employed to develop a wind-power interval estimation model for different meteorological patterns, to better address wind-power uncertainty using historical data; (iii) taking into account the carbon-trading mechanism, a two-stage ARO cooperative operation model based on the improved uncertainty set of wind power is presented to achieve an optimal low-carbon and economically efficient dispatching strategy for MMGSs.

The remainder of this paper is organized as follows. Section 2 presents the interval estimation model of wind power with different meteorological patterns. Section 3 presents the two-stage ARO low-carbon economic dispatching model of MMGS, considering the carbon-trading mechanism. Section 4 introduces the solution method for the presented model. Section 5 presents the experimental analysis. Section 6 presents the conclusions of this paper.

## 2. Improved Data-Driven Uncertainty Set for Wind Power

In this section, based on the limited historical data of wind power and the corresponding multi-dimensional meteorological information, the weather type labels of the original data are obtained using the MRMR feature selection method and CURE clustering algorithm. The CGAN is further employed to generate an appropriate number of wind-power samples with different labels, in order to increase the number of samples for the original wind-power data. Based on the achieved labels and the expanded wind-power data, a grouping generative model of wind power is established. The generated samples are also utilized to construct Wasserstein metric-based ambiguity sets. As a result, the uncertainty interval of wind power can be obtained to precisely describe the uncertainty. The presented data-driven model is illustrated in Figure 1.

### 2.1. Wind Power Generative Model Considering Meteorological Classification

#### 2.1.1. MRMR-Based Feature Selection Method

The features with redundancy or low relevance frequently have negative impacts on the classification of meteorological data. Therefore, MRMR technology is applied to obtain an optimally selected feature subset that has a minimum redundancy among the interior features and maximum relevance to the wind power. The maximum relevance and minimum redundancy indicators are defined in Equations (1) and (2):(1)$$\max D(S,\nu ),D=\frac{1}{\left|S\right|}\sum_{d_{p}\in S}I(d_{p};\nu )$$
(2)$$\min R(S),R=\frac{1}{{\left|S\right|}^{2}}\sum_{d_{p},d_{q}\in S}I(d_{p};d_{q})$$
where *S* denotes the set of the associated features affecting the wind power; *I*(·) represents the mutual information; *D*(*S*, *ν*) describes the relevance between each feature *d_p_* in *S* and the wind power *ν*; *R*(*S*) describes the redundancy among features in *S*; |*S*| denotes the number of features contained in *S*.

Assuming the feature subset *S_k_*_−1_ composed of *k* − 1 features has been determined, the *k*-th feature selected from the remaining feature set should satisfy Equation (3).
(3)$$\underset{d_{j}\in \{S-S_{k-1}\}}{\max}(I(d_{q};\nu )-\frac{1}{k-1}\sum_{d_{p}\in S_{k-1}}I(d_{q};d_{p}))$$

Finally, the meteorological features for which the values of the operator increment are greater than zero are selected to form the feature subset *S_K_*. Then, the selected features and the wind-power data in *S_K_* are normalized using Equation (4). As a result, a normalized dataset ***R***_nor_ containing the optimal feature information and wind power is constructed. The details of ***R***_nor_ are shown in Equation (5).
(4)$$d_{\mathrm{nor}}=\frac{\left(\bar{d}-\underline{d})\cdot (d-d_{\min}\right)}{d_{\max}-d_{\min}}+\underline{d}$$
(5)$$R_{\mathrm{nor}}=\{d_{1,nor},d_{2,nor},\cdots ,d_{K,nor},\nu_{nor}\}$$

Here, $\bar{d}$ and $\underline{d}$ denote the expected maximum and minimum values, which are taken as 1 and −1, respectively; the subscript “nor” indicates that the value has been normalized.

#### 2.1.2. Clustering of Meteorological Features Based on the CURE Algorithm

The CURE clustering algorithm is employed to implement the clustering for the high-dimensional dataset ***R***_met_ to group similar weather patterns. Based on the clustered result, the weather type labels *l*_lab_ can be obtained. Each corresponding wind-power sample in ***R***_nor_ can then be assigned a weather type label. Finally, the dataset ***R***_lab_ containing the labeled wind-power samples can be determined. The compositions of ***R***_met_ and ***R***_lab_ are shown in Equation (6):(6)$$R_{\mathrm{met}}=\{d_{1,nor},d_{2,nor},\cdots ,d_{K,nor}\},R_{\mathrm{lab}}=\{\nu_{\mathrm{nor}},l_{\mathrm{lab}}\}$$

The CURE algorithm is a hierarchical clustering method that uses multiple representative elements to represent a cluster. When clustering the normalized dataset ***R***_met_, each sample is firstly regarded as a cluster. Then, the algorithm combines the two closest clusters to form a new cluster. When the pre-specified number of clusters is reached, the algorithm terminates. The distance between two clusters is the distance between the nearest two representative elements belonging to the two clusters, as determined in accordance with Equation (7). The identification of the representative elements is based on [25].
(7)$$dist(u,w)=\underset{u\prime \in u.\mathrm{rep},w\prime \in w.\mathrm{rep}}{\min}d(u\prime ,w\prime )$$

Here, *u*.rep and *w*.rep represent the collection of the representative elements of cluster *u* and cluster *w*, respectively; *d*(*u*′,*w*′) represents the distance between the representative elements *u*′ and *w*′. The distance is determined using the square of the Euclidean distance.

#### 2.1.3. Generation of Wind-Power Samples Using CGAN

CGAN is an improved model based on the original GAN model; it allows generation of the data samples based on the predefined labels. Therefore, it can be adopted to generate the labeled data samples serving the specific designed scenarios. It trains the model parameters using the adversarial competition between its generator and discriminator. When the Nash equilibrium is achieved, the training process terminates. Then, the samples with given labels can be further generated. Figure 2 illustrates the basic structure of the CGAN model.

The generator and discriminator are both deep neural networks defined by a set of parameters. The objective of training the generator parameters is the generation of newly synthesized samples in which the distribution is similar to the original data and which eventually cheat the discriminator. The objective of training the discriminator parameters is the identification of whether the input sample is a real sample or a synthesized sample. Using the adversarial training of the above two neural networks, we can finally reach the Nash equilibrium point, which indicates that the synthesized samples have sufficiently similar data features to those of the original samples. Therefore, they can be employed to enrich the original to-be-processed dataset.

The training objective function of CGAN is shown in Equation (8). The weather type label *l*_lab_ in the dataset ***R***_lab_ is taken as the conditional information *z* and fed into the generator along with the noise *b*. The generated sample and the real sample *a* are each combined with *z* and then fed into the discriminator. Through the adversarial training of CGAN, the generator can finally generate an appropriate amount of wind-power data as conditioned by the weather type labels to acquire the generated dataset ***R***_gen_, as shown in Equation (9).
(8)$$\underset{G}{\min}\;\underset{D}{\max}V(D,G)=E_{a\sim P_{a}}[\ln D(a|z)]+E_{b\sim P_{b}}[\ln(1-D(G(b|z)))]$$
(9)$$R_{\mathrm{gen}}=\{\nu_{1},\cdots ,\nu_{m},\cdots ,\nu_{M}\}$$

Here, *E*(·) denotes the expected value; *D*(*a*|*z*) is the probability that the real sample *a* conditioned by label *z* is determined to be true in the discriminator. *D*(*G*(*b*|*z*)) is the probability that the generated sample *G*(*b*|*z*) based on the noise *b* and label *z* is determined to be true in the discriminator. $\nu_{m}$ is a generated wind-power sample with label *m*. *M* represents the total number of labels.

### 2.2. Construction of Uncertain Interval under Ambiguity Probability Distribution

The weather type label *l*_lab_ for a certain day can be determined from the meteorological information provided by the weather prediction for the day. Then, the generated samples using the label *l*_lab_ in ***R***_gen_ can be employed to delineate the wind power more precisely. However, the probability distribution of wind power can remain uncertain. Therefore, Wasserstein metric-based ambiguity sets containing a set of uncertain probability distributions are constructed separately for the wind-power scenarios with different labels. Specifically, this involves limitation of the probability distribution ${\tilde{P}}_{m}$ of the true wind power ${\tilde{\nu }}_{m}$ with label *m* in an ambiguity set Ω*_m_*. The probability expectation of the true wind power in the ambiguity set Ω*_m_* with a given uncertainty set is employed as the index to determine the worst-case scenario, as shown in Equation (10):(10)$$\underset{{\tilde{P}}_{m}\in \Omega_{m}}{\sup}E_{P}[P({\tilde{\nu }}_{m}\leq {\hat{\nu }}_{m}^{l}\|{\tilde{\nu }}_{m}\geq {\hat{\nu }}_{m}^{u})]$$
where ${\hat{\nu }}_{m}^{u}$ and ${\hat{\nu }}_{m}^{l}$ represent the upper and lower bound of the uncertainty set of the wind power with label *m*.

The Wasserstein distance is adopted to measure the distance between the empirical distribution ${\hat{P}}_{m0}$ based on the generated sample ${\hat{\nu }}_{m}$ and the true distribution ${\tilde{P}}_{m}$ based on ${\tilde{\nu }}_{m}$, as shown in Equation (10). The ambiguity set is modeled in Equations (11) and (12). The construction of the empirical distribution is based on [17].
(11)$$W({\tilde{P}}_{m},{\hat{P}}_{m0})=\underset{\Pi }{\inf}\left\{\int_{\Xi^{2}}\left\|{\tilde{\nu }}_{m}-{\hat{\nu }}_{m}\right\|\Pi \left(d({\tilde{\nu }}_{m}),d({\hat{\nu }}_{m})\right)\right\}$$
(12)$$\Omega_{m}=\left\{P_{m}\in M(\Xi_{m})|W({\tilde{P}}_{m},{\hat{P}}_{m0})\leq \varepsilon_{m}\right\}$$

Here, ${\hat{\nu }}_{m}$ represents the generated sample with label *m*; ∥⋅∥ represents a norm function, which is *L*^1^-norm in this paper; $\Pi (d({\tilde{\nu }}_{m}),d({\hat{\nu }}_{m}))$ represents the joint probability distribution of ${\hat{\nu }}_{m}$ and ${\tilde{\nu }}_{m}$; $M(\Xi_{m})$ represents the set of all probability distributions of ${\tilde{\nu }}_{m}$ with supporting set $\Xi_{m}$; $\varepsilon_{m}$ controls the size of Ω*_m_*, which depends on the sample number *N_m_* with label *m* and the confidence level of Ω*_m_* [17]. 

We substitute Ω*_m_* into Equation (10) and then convert Equation (10) into the form shown in Equation (13):(13)$$\left\{\begin{array}{l} \underset{\lambda \geq 0,\sigma_{n}}{\inf}\left\{\lambda \varepsilon_{m}+\frac{1}{N_{m}}\sum_{n=1}^{N}\sigma_{n}\right\}\\ s.t.\sigma_{n}=1,\forall {\underline{\nu }}_{m}\leq {\hat{\nu }}_{mn}<{\hat{\nu }}_{m}^{l},{\hat{\nu }}_{m}^{u}<{\hat{\nu }}_{mn}\leq {\bar{\nu }}_{m}\\ \;\;1-\lambda ({\hat{\nu }}_{m}^{u}-{\hat{\nu }}_{n})\leq \sigma_{n},\forall {\hat{\nu }}_{m}^{l}\leq {\hat{\nu }}_{mn}<{\hat{\nu }}_{m}^{u}\\ \;\;1-\lambda ({\hat{\nu }}_{n}-{\hat{\nu }}_{m}^{l})\leq \sigma_{n},\forall {\hat{\nu }}_{m}^{l}\leq {\hat{\nu }}_{mn}<{\hat{\nu }}_{m}^{u}\\ \;\;0\leq \sigma_{n},\forall {\hat{\nu }}_{m}^{l}\leq {\hat{\nu }}_{mn}<{\hat{\nu }}_{m}^{u} \end{array}\right.$$
where λ represents the dual variable; $\sigma_{n}$ represents an auxiliary variable; ${\bar{\nu }}_{m}$ and ${\underline{\nu }}_{m}$ represent the upper limit and the lower limit of ${\tilde{\nu }}_{m}$ in the supporting set $\Xi_{m}$.

In order to simplify the calculation, it is assumed that the boundaries of the uncertainty set, namely ${\hat{\nu }}_{m}^{u}$ and ${\hat{\nu }}_{m}^{l}$, are symmetric about the mean value ${\hat{\mu }}_{m}$, as shown in Equation (14). Therefore, the relationship between the different uncertainty set bounds and different probability expectation values *f_m_* can be obtained using Equation (15):(14)$${\hat{\nu }}_{m}^{l}={\hat{\mu }}_{m}-{\hat{\delta }}_{m},{\hat{\nu }}_{m}^{u}={\hat{\mu }}_{m}+{\hat{\delta }}_{m}$$
(15)$$f_{m}\left({\hat{\delta }}_{m}\right)=\underset{{\tilde{P}}_{m}\in \Omega_{m}}{\sup}E_{P}[P({\tilde{\nu }}_{m}\leq {\hat{\mu }}_{m}-{\hat{\delta }}_{m}||{\tilde{\nu }}_{m}\geq {\hat{\mu }}_{m}+{\hat{\delta }}_{m})]$$
where ${\hat{\delta }}_{m}$ is an auxiliary variable.

The probability point *f_m_* = 1 − *θ* can be approached gradually using the dichotomy denoted by Equation (15). Its corresponding values ${\hat{\nu }}_{m}^{l}$ and ${\hat{\nu }}_{m}^{u}$ are further taken as the lower and the upper boundaries of the uncertainty set *U* to complete the construction of the uncertainty interval with the label *m*.

## 3. Modeling of MMGS Considering the Carbon-Trading Mechanism

The structure of the MMGS studied in this paper is shown in Figure 3. The MMGS consists of a number of *n* combined heat and power microgrids, each of which contains a wind turbine, distribution/gas network, controllable generation (CG) (gas turbine, gas boiler, electric boiler), and electric/thermal load. Each microgrid is connected to the distribution network, while different microgrids are connected to each other via tie line. The dispatch center of every microgrid can exchange limited operational information with other connected microgrids. Therefore, each microgrid has the ability to make decisions independently to interact with the distribution network or the other microgrids in terms of implementing electricity trading. At the same time, the microgrids also conduct carbon trading with the carbon market to optimize the low-carbon and economic performance of their own dispatching strategies.

RO adopts a bounded and closed uncertainty set to describe the variable range of uncertain parameters. Then, the uncertainty set is searched for the case that makes the optimization result the most pessimistic. Therefore, RO is an uncertainty handling method that optimizes for the worst-case scenario in the uncertainty set of uncertain parameters. Based on RO, an adjustable robust parameter is introduced to control the frequency of the worst-case scenario in the uncertainty set. Thus, the ARO model is obtained.

In order to effectively cope with the risk brought by the uncertainty of wind-turbine output to the microgrid dispatching, we construct a two-stage ARO microgrid model using the following compact form, as shown in Equation (16):(16)$$\underset{x}{\min}\{C^{\mathrm{da}}(x)+\underset{u}{\max}\;\underset{y}{\min}\;C^{\mathrm{rt}}(u,y)\}$$
where ***x***, ***u***, ***y***, *C*^da^(***x***), and *C*^rt^(***u***,***y***) represent the day-ahead dispatching strategy, the worst-case scenario of wind-power output, the real-time adjustment strategy for ***x*** with the worst-case scenario, the day-ahead operation cost, and the real-time adjustment cost, respectively.

The uncertainty set $U=\{{\tilde{\nu }}_{\mathrm{WT}}\in [\nu_{\mathrm{WT}}^{l},\nu_{\mathrm{WT}}^{u}]\}$ is adopted to describe the uncertainty of wind power in the adjustment stage. In the aforementioned presentation, an adjustable robust parameter represents the number of occurrences of the worst-case scenarios of wind-power output during the dispatch period. The adjustable robust parameter *Γ* is introduced to control the conservativeness of the model. Therefore, the wind-power output can be described as shown in Equation (17):(17)$$\sum_{t\in T}\left(\frac{{\tilde{P}}_{\mathrm{WT},t}-P_{\mathrm{WT},t}^{\mathrm{pre}}}{\nu_{\mathrm{WT},t}^{u}}\rho_{t}^{+}+\frac{P_{\mathrm{WT},t}^{\mathrm{pre}}-{\tilde{P}}_{\mathrm{WT},t}}{\nu_{\mathrm{WT},t}^{l}}\rho_{t}^{-}\right)\leq \Gamma$$
where $\nu_{\mathrm{WT},t}^{u}$ and $\nu_{\mathrm{WT},t}^{l}$ represent the fluctuation ranges of wind power at time *t*, which is explained in detail in Section 2; $P_{\mathrm{wind},t}^{\mathrm{pre}}$ and ${\tilde{P}}_{\mathrm{wind},t}$ represent the predicted value and real value of wind-power output at time *t*, respectively; $\rho_{t}^{+}$ and $\rho_{t}^{-}$ represent auxiliary 0–1 variables; and *T* represents the time set.

### 3.1. Modeling of Microgrid Dispatch in the First Stage

In the first stage, Microgrid *i* formulates the dispatching strategy ***x*** with the objective of minimizing the day-ahead operation costs, which include the cost of purchasing and selling electricity within the distribution network $C_{\mathrm{Grid},i}^{\mathrm{da}}$, the cost of purchasing gas from the gas network $C_{\mathrm{Gas},i}^{\mathrm{da}}$, the cost of carbon trading $C_{C,i}^{\mathrm{da}}$, and the cost of trading electricity with other microgrids *C*_MG,*i*_. The details are shown in Equations (18)–(21).
(18)$$\underset{x}{\min}C^{\mathrm{da}}(x)=C_{\mathrm{Grid},i}^{\mathrm{da}}+C_{\mathrm{Gas},it}^{\mathrm{da}}+C_{C,i}^{da}+C_{\mathrm{MG},i}$$
(19)$$C_{\mathrm{Grid},i}^{\mathrm{da}}=\sum_{t\in T}\left(\tau_{\mathrm{ebuy},t}^{\mathrm{da}}P_{\mathrm{ebuy},i,t}^{\mathrm{da}}-\tau_{\mathrm{esell},t}^{\mathrm{da}}P_{\mathrm{esell},i,t}^{\mathrm{da}}\right)$$
(20)$$C_{\mathrm{Gas},i}^{\mathrm{da}}=\sum_{t\in T}\tau_{\mathrm{Gas}}(G_{\mathrm{GT},i,t}^{\mathrm{da}}+G_{\mathrm{GB},i,t}^{\mathrm{da}})$$
(21)$$C_{\mathrm{MG},i}=\sum_{t\in T}\sum_{j\in \Psi_{i}}\tau_{\mathrm{ex},t}P_{ij,t}^{\mathrm{ex}}$$

Here, $\tau_{\mathrm{ebuy},t}^{\mathrm{da}}$, $\tau_{\mathrm{esell},t}^{\mathrm{da}}$, $\tau_{\mathrm{Gas}}$, and $\tau_{\mathrm{ex},t}$ represent the price of electricity purchased from the distribution network, the price of electricity sold to the distribution network, the price of gas purchased from the gas network, and the price of electricity in interactions with other microgrids at time *t* in the day-ahead stage; $\Psi_{i}$ represents the set of microgrids connected to Microgrid *i*; $P_{\mathrm{ebuy},i,t}^{\mathrm{da}}$, $P_{\mathrm{esell},i,t}^{\mathrm{da}}$, $G_{\mathrm{GT},i,t}^{\mathrm{da}}$, $P_{\mathrm{GB},i,t}^{\mathrm{da}}$, and $P_{ij,t}^{\mathrm{ex}}$ represent the amount of electricity purchased from the distribution network, the amount of electricity sold to the distribution network, the amount of gas consumed by the gas turbine, the amount of gas consumed by the gas boiler, and the amount of electricity interacting with Microgrid *j* in the day-ahead stage at time *t* of Microgrid *i*, respectively. The details for the cost of carbon trading are provided in Section 3.3.

The constraints in the first stage include energy-balance constraints, constraints for interaction with distribution/gas networks, constraints for power interaction with other microgrids, CG operation constraints, and carbon-balance constraints. The details of the constraints are shown in Equations (22)–(27).

A.Energy balance constraints:(22)$$P_{\mathrm{GT},i,t}^{\mathrm{da}}+P_{\mathrm{Wind},i,t}^{\mathrm{pre}}+P_{\mathrm{ebuy},i,t}^{\mathrm{da}}+\sum_{j\in \Psi_{i}}P_{ij,t}^{\mathrm{ex}}=P_{\mathrm{EB},i,t}^{\mathrm{da}}+P_{\mathrm{esel},i,t}^{\mathrm{da}}+P_{\mathrm{Eload},i,t}$$
(23)$$Q_{\mathrm{GT},i,t}^{\mathrm{da}}+Q_{\mathrm{GB},i,t}^{\mathrm{da}}+Q_{\mathrm{EB},i,t}^{\mathrm{da}}=P_{\mathrm{Hload},i,t}$$
where $P_{\mathrm{GT},i,t}^{\mathrm{da}}$, $P_{\mathrm{EB},i,t}^{\mathrm{da}}$, $P_{\mathrm{Eload},i,t}$, $Q_{\mathrm{EB},i,t}^{\mathrm{da}}$, and $P_{\mathrm{Hload},i,t}$ represent the electric output of the gas turbine, the electric power consumed by the electric boiler, the electric load demand, the thermal output of the electric boiler, and the thermal load demand in Microgrid *i* at time *t* in the day-ahead stage, respectively.B.Constraints for interaction with distribution/gas networks:(24)$$\left\{\begin{array}{l} 0\leq P_{\mathrm{ebuy},i,t}^{\mathrm{da}}\leq V_{\mathrm{ebuy},i,t}P_{\mathrm{Grid},i}^{\max}\\ 0\leq P_{\mathrm{esell},i,t}^{\mathrm{da}}\leq V_{\mathrm{esell},i,t}P_{\mathrm{Grid},i}^{\max}\\ V_{\mathrm{ebuy},i,t}+V_{\mathrm{esell},i,t}\leq 1\\ 0\leq G_{\mathrm{GT},i,t}^{\mathrm{da}}+G_{\mathrm{GB},i,t}^{\mathrm{da}}\leq G_{\mathrm{Gas},i}^{\max} \end{array}\right.$$
where *V*_ebuy,*i,t*_ and *V*_esell,*i,t*_ represent the states of the power purchased and sold from Microgrid *i* to the distribution network at time t, which are 0–1 variables; $P_{\mathrm{Grid},i}^{\max}$ and $P_{\mathrm{Gas},i}^{\max}$ represent the maximum value of power interacting with the distribution network and the maximum value of gas purchased from the gas network, respectively.C.Constraints for power interaction with other microgrids:(25)$$-P_{ij}^{\max}\leq P_{ij,t}^{\mathrm{ex}}\leq P_{ij}^{\max}$$
where $P_{ij}^{\max}$ represents the maximum value of the transaction power between Microgrid *i* and Microgrid *j*.D.CG operation constraints:(26)$$\left\{\begin{array}{c} V_{\mathrm{CG},i,t}P_{\mathrm{CG},i}^{\min}\leq P_{\mathrm{CG},i,t}^{\mathrm{da}}\leq V_{\mathrm{CG},i,t}P_{\mathrm{CG},i}^{\max}\\ -R_{\mathrm{CG},i}^{\mathrm{dw}}\leq P_{\mathrm{CG},i,t}^{\mathrm{da}}-P_{\mathrm{CG},i,t-1}^{\mathrm{da}}\leq R_{\mathrm{CG},i}^{\mathrm{up}} \end{array}\right.$$
where *V*_CG,*i,t*_ represents the operation state of the CG, which is a 0–1 variable; $P_{\mathrm{CG},i}^{\min/\max}$ and $R_{\mathrm{CG},i}^{\mathrm{up}/\mathrm{dw}}$ represent the maximum/minimum output of the CG and the maximum up/down ramping capacities, respectively.

Energy conversion model for CG:(27)$$\left\{\begin{array}{l} P_{\mathrm{GT},i,t}^{\mathrm{da}}=\eta_{\mathrm{GT}}H_{\mathrm{ng}}G_{\mathrm{GT},i,t}^{\mathrm{da}}\\ Q_{\mathrm{GT},i,t}^{\mathrm{da}}=\eta_{\mathrm{HE}}\eta_{\mathrm{WH}}(1-\eta_{GT})H_{\mathrm{ng}}G_{\mathrm{GT},i,t}^{\mathrm{da}}\\ Q_{\mathrm{GB},i,t}^{\mathrm{da}}=\eta_{\mathrm{GB}}H_{ng}G_{\mathrm{GB},i,t}^{\mathrm{da}}\\ Q_{\mathrm{EB},i,t}^{\mathrm{da}}=\eta_{\mathrm{EB}}P_{\mathrm{EB},i,t}^{\mathrm{da}} \end{array}\right.$$
where *η*_GT_, *η*_WH_, *η*_HE_, *η*_GB_, and *η*_EB_ represent the conversion efficiencies of the gas turbine power generation, the waste heat recovery, the heat exchanger, the gas boiler heating, and the electric boiler heating, respectively. *H*_ng_ represents the heat value of natural gas.

### 3.2. Modeling Real-Time Adjustment Dispatch in the Second Stage

Based on the ***x*** obtained in the first stage, Microgrid *i* searches for the worst-case scenario of wind power ***u*** and formulates the corresponding real-time adjustment strategy ***y*** after the realization of the wind-power uncertainty. The real-time adjustment cost consists of the regulation cost of CG $C_{\mathrm{CG},i}^{\mathrm{rt}}$, the real-time grid interaction cost $C_{\mathrm{Grid},i}^{\mathrm{rt}}$, the real-time gas purchasing cost $C_{\mathrm{Gas},i}^{\mathrm{rt}}$, the real-time carbon trading cost $C_{C,i}^{\mathrm{rt}}$, and the wind curtailment cost $C_{\mathrm{Wind},i}^{\mathrm{loss}}$. The details are shown in Equations (28)–(32).
(28)$$C^{\mathrm{rt}}(u,y)=C_{\mathrm{CG},i}^{\mathrm{rt}}+C_{\mathrm{Grid},i}^{\mathrm{rt}}+C_{\mathrm{Gas},i}^{\mathrm{rt}}+C_{C,i}^{rt}+C_{\mathrm{Wind},i}^{\mathrm{loss}}$$
(29)$$C_{\mathrm{CG},i}^{\mathrm{rt}}=\sum_{t\in T}(\tau_{\mathrm{GT}}^{\mathrm{up}}G_{\mathrm{GT},i,t}^{\mathrm{up}}+\tau_{\mathrm{GT}}^{\mathrm{dw}}G_{\mathrm{GT},i,t}^{\mathrm{dw}}+\tau_{\mathrm{GB}}^{\mathrm{up}}G_{\mathrm{GB},i,t}^{\mathrm{up}}+\tau_{\mathrm{GB}}^{\mathrm{dw}}G_{\mathrm{GB},i,t}^{\mathrm{dw}}+\tau_{\mathrm{EB}}^{\mathrm{up}}P_{\mathrm{EB},i,t}^{\mathrm{up}}+\tau_{\mathrm{EB}}^{\mathrm{dw}}P_{\mathrm{EB},i,t}^{\mathrm{dw}})$$
(30)$$C_{\mathrm{Grid},i}^{\mathrm{rt}}=\sum_{t\in T}\left(\tau_{\mathrm{ebuy},t}^{\mathrm{rt}}P_{\mathrm{ebuy},i,t}^{\mathrm{rt}}-\tau_{\mathrm{esell},t}^{\mathrm{rt}}P_{\mathrm{esell},i,t}^{\mathrm{rt}}\right)$$
(31)$$C_{\mathrm{Gas},i}^{\mathrm{rt}}=\sum_{t\in T}\tau_{\mathrm{Gas}}(G_{\mathrm{GT},i,t}^{\mathrm{up}}-G_{\mathrm{GT},i,t}^{\mathrm{dw}}+G_{\mathrm{GB},i,t}^{\mathrm{up}}-G_{\mathrm{GB},i,t}^{\mathrm{dw}})$$
(32)$$C_{\mathrm{Wind},i}^{\mathrm{loss}}=\sum_{t}\tau_{\mathrm{loss}}({\tilde{P}}_{\mathrm{Wind},i,t}-P_{\mathrm{Wind},i,t}^{\mathrm{get}})$$

Here, $\tau_{\mathrm{GT}}^{\mathrm{up}/\mathrm{dw}}$, $\tau_{\mathrm{GB}}^{\mathrm{up}/\mathrm{dw}}$, $\tau_{\mathrm{EB}}^{\mathrm{up}/\mathrm{dw}}$, and $\tau_{\mathrm{loss}}$ represent the up/down regulation of the price of the GT, the up/down regulation of the price of the GB, the up/down regulation of the price of the EB, and the wind penalty price, respectively; $\tau_{\mathrm{ebuy},t}^{\mathrm{rt}}$ and $\tau_{\mathrm{esell},t}^{\mathrm{rt}}$ represent the purchased and sold prices of electricity in the real-time stage, respectively; $G_{\mathrm{GT},i,t}^{\mathrm{up}/\mathrm{dw}}$, $G_{\mathrm{GB},i,t}^{\mathrm{up}/\mathrm{dw}}$, and $P_{\mathrm{EB},i,t}^{\mathrm{up}/\mathrm{dw}}$ represent the up/down regulation of gas to the GT, the up/down regulation of gas to the GB, and the up/down regulation of power to the EB, respectively; $P_{\mathrm{ebuy},i,t}^{\mathrm{rt}}$ and $P_{\mathrm{esell},i,t}^{\mathrm{rt}}$ represent the purchased and sold power in the real-time stage, respectively; $P_{\mathrm{wind},i,t}^{\mathrm{get}}$ represents the real-time injected wind power.

The constraints in the second stage consist of the wind turbine regulation constraints, the CG regulation constraints, the real-time energy balance constraints, and the real-time power/gas interaction constraints. The details of the constraints are shown in Equations (33)–(37).

A.Wind turbine regulation constraints:(33)$$0\leq P_{\mathrm{Wind},i,t}^{\mathrm{get}}\leq {\tilde{P}}_{\mathrm{Wind},i,t}$$B.CG regulation constraints:(34)$$\left\{\begin{array}{l} 0\leq P_{\mathrm{CG},i,t}^{\mathrm{up}}\leq V_{\mathrm{CG},i,t}^{\mathrm{up}}R_{\mathrm{CG},i}^{\mathrm{up}},0\leq P_{\mathrm{CG},i,t}^{\mathrm{dw}}\leq V_{\mathrm{CG},i,t}^{\mathrm{dw}}R_{\mathrm{CG},i}^{\mathrm{dw}}\\ V_{\mathrm{CG},i,t}^{\mathrm{up}}+V_{\mathrm{CG},i,t}^{\mathrm{dw}}\leq 1\\ V_{\mathrm{CG},i,t}P_{\mathrm{CG},i}^{\min}\leq P_{\mathrm{CG},i,t}^{\mathrm{da}}+P_{\mathrm{CG},i,t}^{\mathrm{up}}-P_{\mathrm{CG},i,t}^{\mathrm{dw}}\leq V_{\mathrm{CG},i,t}P_{\mathrm{CG},i}^{\max}\\ -R_{\mathrm{CG},i}^{\mathrm{dw}}\leq P_{\mathrm{CG},i,t}^{\mathrm{da}}+P_{\mathrm{CG},i,t}^{\mathrm{up}}-P_{\mathrm{CG},i,t}^{\mathrm{dw}}-(P_{\mathrm{CG},i,t-1}^{\mathrm{da}}+P_{\mathrm{CG},i,t-1}^{\mathrm{up}}-P_{\mathrm{CG},i,t-1}^{\mathrm{dw}})\leq R_{\mathrm{CG},i}^{\mathrm{up}} \end{array}\right.$$
where $V_{\mathrm{CG},i,t}^{\mathrm{up}/\mathrm{dw}}$ represents the ramping state of the CG, which is a 0-1 variable.C.Real-time energy balance constraints:(35)$$\begin{array}{l} P_{\mathrm{GT},i,t}^{\mathrm{da}}+P_{\mathrm{GT},i,t}^{\mathrm{up}}-P_{\mathrm{GT},i,t}^{\mathrm{dw}}+P_{\mathrm{Wind},i,t}^{\mathrm{get}}+P_{\mathrm{ebuy},i,t}^{\mathrm{da}}+P_{\mathrm{ebuy},i,t}^{\mathrm{rt}}+\sum_{j\in \Psi_{i}}P_{ij,t}^{\mathrm{ex}}\\ \;=P_{\mathrm{EB},i,t}^{\mathrm{da}}+P_{\mathrm{EB},i,t}^{\mathrm{up}}-P_{\mathrm{EB},i,t}^{\mathrm{dw}}+P_{\mathrm{esell},i,t}^{\mathrm{da}}+P_{\mathrm{esell},i,t}^{\mathrm{rt}}+P_{\mathrm{Eload},i,t} \end{array}$$
(36)$$Q_{\mathrm{GT},i,t}^{\mathrm{da}}+Q_{\mathrm{GT},i,t}^{\mathrm{up}}-Q_{\mathrm{GT},i,t}^{\mathrm{dw}}+Q_{\mathrm{GB},i,t}^{\mathrm{da}}+Q_{\mathrm{GB},i,t}^{\mathrm{up}}-Q_{\mathrm{GB},i,t}^{\mathrm{dw}}+Q_{\mathrm{EB},i,t}^{\mathrm{da}}+Q_{\mathrm{EB},i,t}^{\mathrm{up}}-Q_{\mathrm{EB},i,t}^{\mathrm{dw}}=P_{\mathrm{Hload},i,t}$$D.Real-time power/gas interaction constraints:(37)$$\left\{\begin{array}{l} 0\leq P_{\mathrm{ebuy},i,t}^{\mathrm{rt}}\leq P_{\mathrm{Grid},i}^{\max}\\ 0\leq P_{\mathrm{esell},i,t}^{\mathrm{rt}}\leq P_{\mathrm{Grid},i}^{\max}\\ 0\leq P_{\mathrm{ebuy},i,t}^{\mathrm{da}}+P_{\mathrm{ebuy},i,t}^{\mathrm{rt}}\leq V_{\mathrm{ebuy},i,t}P_{\mathrm{Grid},i}^{\max}\\ 0\leq P_{\mathrm{esell},i,t}^{\mathrm{da}}+P_{\mathrm{esell},i,t}^{\mathrm{rt}}\leq V_{\mathrm{esell},i,t}P_{\mathrm{Grid},i}^{\max}\\ 0\leq G_{\mathrm{GT},i,t}^{\mathrm{da}}+G_{\mathrm{GT},i,t}^{\mathrm{up}}-G_{\mathrm{GT},i,t}^{\mathrm{dw}}\\ +G_{\mathrm{GB},i,t}^{\mathrm{da}}+G_{\mathrm{GB},i,t}^{\mathrm{up}}-G_{\mathrm{GB},i,t}^{\mathrm{dw}}\leq G_{\mathrm{Gas},i}^{\max} \end{array}\right.$$

### 3.3. Carbon-Trading Mechanism

When there is a shortage of carbon allowance for a microgrid, it must buy its carbon allowance from the carbon market. The cost of purchasing the carbon allowance should be regarded as a part of the operation costs. In contrast, the microgrid can sell the surplus carbon allowance to receive certain monetary compensation. The introduction of the carbon-trading cost will change the total operational cost of the microgrid, thus guiding the microgrid to adopt an efficient dispatching strategy to achieve the goal of reducing carbon emissions. The microgrid employs the stepped carbon-trading mechanism to conduct carbon trading with the carbon market, as shown in Equation (38).
(38)$$C_{C,i}=\left\{\begin{array}{l} \tau_{C}E_{i},E_{i}\leq l_{C}\\ \tau_{C}(1+\rho )(E_{i}-l_{C})+\tau_{C}l_{C},l_{C}\leq E_{i}\leq 2l_{C}\\ \tau_{C}(1+2\rho )(E_{i}-2l_{C})+(2+\rho )\tau_{C}l_{C},2l_{C}\leq E_{i}\leq 3l_{C}\\ \tau_{C}(1+3\rho )(E_{i}-3l_{C})+(3+3\rho )\tau_{C}l_{C},3l_{C}\leq E_{i}\leq 4l_{C}\\ \tau_{C}(1+4\rho )(E_{i}-4l_{C})+(4+6\rho )\tau_{C}l_{C},4l_{C}\leq E_{i} \end{array}\right.$$

Here, $C_{C,i}$, $\tau_{C}$, $l_{C}$, ρ, and *E_i_* represent the carbon trading cost, the initial carbon-trading price, the interval length of the carbon emission, the increase ratio of the stepped price, and the carbon emissions traded with the carbon market, respectively.

Carbon emissions of the microgrid are mainly caused by the power purchased from the distribution network, the operation of the gas turbine, and the operation of the gas boiler. In Microgrid *i*, the formulation of the actual carbon emissions *E_i_*_,a_ and the free allowance for the carbon emission *E_i_*_,0_ are presented in [24] in detail. The carbon-balance constraint in the dispatching process of Microgrid *i* is shown in Equation (39):(39)$$E_{i,a}-E_{i,0}=E_{i}$$

## 4. Model Solution

Since each microgrid belongs to a different party, in order to guarantee the privacy of the information and the independence of the decision making of each microgrid, ADMM is employed to achieve a decentralized solution for MMGS optimal dispatching. At first, the tie-line transmission power between two microgrids is decoupled using the coordination variables ${\hat{P}}_{ij,t}$ and the corresponding consistency constraint shown in Equation (40). Then, the augmented Lagrangian function shown in Equation (41) is constructed. Therefore, the original problem can be decomposed into multiple local optimization problems, which can be solved in each individual microgrid system.
(40)$$P_{ij,t}^{\mathrm{ex}}={\hat{P}}_{ij,t}$$
(41)$$\left.\underset{x}{\min}\left\{c^{T}x+\sum_{t\in T}\sum_{j\in \Psi_{i}}\left[\lambda_{ij,t}^{q}\left(P_{ij,t}^{\mathrm{ex},q}-{\hat{P}}_{ij,t}^{q}\right)+\right.\right.\left.\frac{\rho_{\mathrm{pen}}}{2}{\left\|P_{ij,t}^{\mathrm{ex},q}-{\hat{P}}_{ij,t}^{q}\right\|}_{2}^{2}\right]+\underset{u}{\max}\underset{y}{\min}\left(d^{T}y+e^{T}u\right)\right\}$$

Here, *λ_ij_*_,*t*_ represents the dual variable; $\rho_{\mathrm{pen}}$ represents the penalty factor, which is greater than zero; the superscript *q* represents the number of iterations.

Each microgrid solves the optimization problem locally to obtain the local dispatching strategies and the tie-line coupling variables. The two-stage ARO problem can be solved using the C&CG algorithm. The details of the C&CG algorithm are presented in the Appendix A [11]. The original problem can be decomposed into a master problem and a sub problem. Then, the sub and master problems iterate alternately. The detailed procedure of the nested iterative solution of ADMM and C&CG algorithms is shown in Algorithm 1.
**Algorithm 1** The nested iterative solution process of the ADMM and C&CG methods  1: Initialize: dual variable *λ*, coordination variables ${\hat{P}}_{ij,t}$, convergence gap *ξ*, number of iterations *q*  2: Calculate: initial power of tie-line  3: Update: dual variable $\lambda^{q}$ and coordination variables ${\hat{P}}_{ij,t}^{q}$  4: Each microgrid solves the optimal economic dispatching model in a decentralized way  5: Initialize: upper boundary *U*_0_, lower boundary *L*_0_, initial scenario set *u*_1_, convergence gap ε, number of iterations *l* for robust dispatching  6: Calculate: master problem  7: Obtain: day-ahead dispatching strategy and corresponding adjustment strategy (*x_l_*,*y_l_*)  8: Update: lower boundary *L_l_*  9: Calculate: sub problem10: Obtain: worst-case wind-power scenarios and corresponding adjustment strategy ($x_{l}^{*}$,$y_{l}^{*}$)11: Update: upper boundary *U_l_*12: Repeat Step 6 to Step 11 until $U_{l}-L_{l}<\varepsilon$13: Output: economic dispatching strategy for each microgrid14: Update: dual variable $\lambda^{q+1}$, coordination variable ${\hat{P}}_{ij,t}^{q+1}$15: Repeat Step 4 to Step 14 until the primary residuals and dual residuals are less than the specified convergence gap *ξ*

## 5. Case Study

### 5.1. Basic Parameter Setting

In order to verify the effectiveness of the presented data-driven wind-power uncertainty-set-based low-carbon economic dispatching model for MMGSs, experiments are conducted in the Win10 64-bit system. The optimal dispatching operation of the MMGS is modeled using MATLAB R2018b and solved by Gurobi. The deep-learning network is built in the Keras framework [26] based on Tensorflow 2.2, Python 3.6. An MMGS composed of four microgrids is employed in this paper. The architecture of the MMGS is shown in Figure 3. The structure and device parameters of the four microgrids are identical, while the electric and thermal load demands and wind-power outputs of the microgrids are different. The details of the load curves are shown in Figure 4. The system device parameters are shown in Table 1. The operation cost coefficients are shown in Table 2. The prices of electricity purchased and sold from the distribution network are shown in Table 3. 

In terms of the CGAN employed in this paper, the generator adopts four fully connected layers. The number of neurons in the hidden layers is 128, 256, and 128. The activation function is a rectified linear unit (ReLU). In addition, the size of the output layer is set to 24, which is the same as the number of real daily wind-power samples. In the discriminator, four fully connected layers are also adopted. The number of neurons in the hidden layers is 256, 128, and 64. The activation function is also ReLU. The number of neurons in the output layer is set to 1. The historical data of a wind farm from 2017 to 2018 (640 days in total) are employed. The dataset includes daily multi-dimensional meteorological data and wind-power data, with a sampling interval of 1 h. Additionally, the historical data of 500 days are selected as the training dataset and the historical data of the remaining 140 days are the testing dataset.

### 5.2. Validity Analysis of Uncertainty Set

Due to the large variability of the meteorological characteristics in different seasons, the data of four days from four seasons are randomly selected from the testing dataset to illustrate the capability of the presented model in accurately describing the wind power. The data of these four days are substituted into the presented generative model with labels in this paper (Model 1) and the generative model without feature selection and feature clustering (Model 2) for comparison. The number of samples generated by these two generative models is set to 3000, and the confidence level is 90%. Figure 5 shows the uncertainty intervals of the four-day data with the two models. In the figure, the deep blue line, the orange interval, and the gray interval are the actual value of the wind power, the uncertainty interval of Model 1, and the uncertainty interval of Model 2, respectively.

Figure 5 indicates that in the four-day experiments, the widths of the uncertain intervals of the wind power for Model 1 and Model 2 are similar. However, the interval of Model 1 can fully cover the actual value of the wind power, while that of Model 2 can only partially cover the value. When the width of the uncertainty interval is close to the actual value, the coverage rate is higher and the uncertainty portrayal is more accurate. The method presented in this paper can achieve a higher accuracy in describing wind power for different meteorological patterns in all four seasons.

In order to evaluate the performance of CGAN in enriching the dataset, the samples of the training dataset and the noise are inputted into CGAN to generate 3000 synthesized samples. Then, the probability distribution of the generated samples is compared to the distribution of the real samples. The comparison results of the sample distribution with Label 1 are shown in Figure 6. It can be observed that the distribution results of the generated samples based on the CGAN model are basically consistent with the real samples, demonstrating the effectiveness of CGAN in sample generation. 

In order to verify the effectiveness of the data-driven ambiguity set, the widths of the wind-power uncertainty interval with different sample sizes are compared. The comparison results with Label 1 are shown in Figure 7. As the number of samples utilized to construct the ambiguity set increases, the width of the obtained uncertainty interval gradually narrows. This is mainly because the Wasserstein ball can limit the fluctuation range of the probability distribution of wind power. The radius of the ambiguity set decreases with the increasing number of samples, resulting in the unknown uncertainty distribution approximating the true distribution of the historical data. Therefore, the conservativeness of the model can be reduced. This indicates that the interval estimation using the data-driven model presented in this paper, i.e., the expansion of the number of wind-power samples serving the construction of the Wasserstein distance-based ambiguity sets, can effectively improve the accuracy of the uncertainty portrayal when the historical data are insufficient.

### 5.3. Analysis of Data-Driven Microgrid Dispatching Results

#### 5.3.1. Comparison of Dispatching Results with Different Sample Sizes

In order to analyze the impact of the sample size on the microgrid dispatching, the uncertainty intervals of the wind power driven by the sample sizes of the original amount (102), 300, 500, 1000, and 3000 with Label 1 are employed for the dispatching of Microgrid 1. The comparison of the cost and the carbon emission is shown in Figure 8. It can be observed that with the increasing number of samples, both the cost and the carbon emissions of the microgrid dispatching decrease. This is because more samples drive the uncertainty interval to become narrower. As a result, the optimization can guarantee dispatching under the worst scenario with a higher efficiency. Meanwhile, wind-power consumption and carbon-emission reduction also benefit from the improvement in the accuracy of the wind-power description. The results significantly suggest that data-driven wind-power interval estimation can effectively improve efficiency and reduce the carbon emissions of microgrid dispatching.

#### 5.3.2. Comparison of Dispatching Results of Uncertainty Optimization Algorithms

Subsequent experiments compared the performance of the presented data-driven ARO method to that of the traditional RO and ARO methods in terms of microgrid dispatching. In the experiment, the prediction errors of the wind-power uncertainty interval of the traditional RO and ARO methods are assumed to be 20%. The presented data-driven ARO adopts the uncertainty interval using 3000 generated samples with Label 4. The experiment is carried out based on the dispatching of Microgrid 2. The results are shown in Figure 9. The calculation times of the three optimization methods are shown in Table 4.

The RO method utilizes the worst-case scenarios of wind power to achieve optimal dispatching, which results in sacrificing both efficiency and lower carbon emissions to improve robustness. Consequently, the results are conservative compared to the other optimization methods. The ARO method introduces adjustable robust parameters to limit the appearance frequency of the worst-case scenarios, making its dispatching results more efficient and less carbon-emitting than those of RO. However, since the distribution information of the uncertain variables is neglected in ARO, the results remain conservative compared to those of the presented data-driven ARO, from which a more accurate uncertainty boundary can be obtained. Thus, a balance between the efficiency and robustness of the dispatching strategy can be achieved, and carbon emissions can be further reduced. 

Table 4 shows that the traditional RO takes the shortest computation time, while the traditional ARO and data-driven ARO require a longer solution time due to multiple iterations. The solution efficiencies of traditional ARO and data-driven ARO are similar. This indicates that the improved ARO model presented in this paper does not carry a burden on computational performance based on the traditional ARO.

### 5.4. Analysis of MMGS Dispatching with Different Dispatching Cases

#### 5.4.1. Comparison of Dispatching Results Using Different Dispatching Cases

The four dispatching cases shown in Table 5 are conducted to verify the effectiveness of the MMGS dispatching model presented in this paper.

Figure 10 shows the results of MMGS dispatching in four different cases. It can be observed that without the energy interaction and carbon trading, the dispatching strategies of the MMGS generate greater costs and carbon emissions. Although the introduction of carbon trading generates a slightly higher cost, it significantly reduces carbon emissions. The results suggest that involving energy interaction and carbon trading in the dispatching of an MMGS is an effective way to improve the efficiency and lower the carbon emissions of the system. The detailed data of the experiment are shown in Table 6, which gives the energy-interaction costs with external networks (distribution network and gas network), energy-interaction costs among microgrids, device-dispatching costs, carbon-trading costs, and carbon emissions of the four microgrids. 

Compared to Case 2, the energy-interaction cost of Case 4 is significantly changed. The energy-purchase costs of the power-deficit Microgrids 1 and 2 decrease by USD 428.5167 and 392.7376, respectively. Simultaneously, the energy-interaction costs between the two microgrids increase by USD 250.142 and USD 246.2674, respectively. Accordingly, the energy-sold revenues due to the power surplus in Microgrids 3 and 4 decrease by USD 212.4599 and USD 254.3293, respectively. The energy-interaction revenues between the two microgrids increase by USD 219.9794 and USD 277.4536, respectively. The total cost of the MMGS decreases by USD 661.8415. The comparison between Case 3 and Case 1 also shows similar results. This is because the power interaction among microgrids provides an additional energy interaction avenue for the microgrids. As the interaction price is between the purchased and sold prices from the distribution network, the microgrid with surplus electricity can sell electricity with a higher price while the microgrid with power shortage can purchase power with a lower price. Each microgrid is able to change its energy-exchanging strategy; this method favors power interaction among microgrids as it reduces the energy-purchase cost and increases the energy-sold revenue. Eventually, the total cost of the MMGS can be lower.

Comparing the MMGS dispatching results before and after considering the carbon-trading mechanism, it can be observed that due to the introduction of the carbon trading in Cases 2 and 4, the total dispatching cost of each microgrid increases compared to that of Cases 1 and 3. The total cost of the MMGS increases by USD 458.2795 and USD 76.5478, respectively. However, the carbon emissions of most microgrids are decreased. The total carbon emission of the MMGS decreases by 3641.78 kg and 752.78 kg, respectively. As purchasing electricity from the distribution network and the output of gas units both generate carbon emissions, the consideration of the carbon costs of the microgrids would increase their total dispatching costs. However, after the introduction of the carbon-trading mechanism, the microgrids prefer using relatively low-carbon gas units and carbon-free wind power to meet load demands over purchasing electricity from the distribution network. The microgrids are able to strategically choose energy-supply methods to obtain low costs and achieve the reduction in carbon emissions. This illustrates the effectiveness of the carbon-trading mechanism in guiding MMGSs to reduce their carbon emissions.

#### 5.4.2. Analysis of Power Interaction among Microgrids

Figure 11 shows the results of the electricity interaction of each microgrid in Case 4. The positive step curve “MG1-MG2” in the figure indicates that Microgrid 1 buys electricity from Microgrid 2, while the negative one indicates that Microgrid 1 sells electricity to Microgrid 2.

As can be observed from Figure 11, the amount of electricity exchanged among microgrids is low during the periods of 01:00–06:00 and 23:00–24:00. This is because the loads of all four microgrids are at low levels during these periods. Wind power is able to meet their individual load demand, and they do not require a large amount of energy interaction with the other microgrids. In other periods, the load demands of Microgrids 1 and 2 increase, resulting in the wind power in the microgrids not being sufficient to balance the loads. At the same time, the wind power of Microgrids 3 and 4 is surplus; they have more than enough power to meet their own load demand. As a result, there is a large amount of energy interaction between Microgrids 1 and 2 and Microgrids 3 and 4.

## 6. Conclusions

In order to ensure efficient low-carbon dispatching in the power system, a data-driven wind-power uncertainty-set-based low-carbon optimization dispatching model considering wind-power uncertainty for MMGSs is presented here. The main conclusions drawn from the case studies are as follows:Synthesized wind-power samples can be generated to enrich the number of samples using CGAN. Then, based on the meteorological classification, the grouped ambiguity sets are constructed, and the wind-power uncertainty intervals can be obtained. Compared to the interval estimation model without sample generation and meteorological classification, this method can reduce the width of the uncertain intervals, which improves the accuracy of the wind-power description.An ARO model based on the improved wind-power uncertainty intervals is further constructed. Compared to the traditional uncertainty optimization methods, the presented ARO model can improve wind-power consumption by decreasing the deviation between the real wind power and the uncertainty intervals. Therefore, based on the dispatching optimization of the ARO model, the efficiency can be improved and carbon emissions can be reduced.Additionally, a low-carbon dispatching strategy considering power interaction among microgrids is further presented. This strategy can guarantee the benefits of each microgrid while reducing the carbon emissions as a result of microgrid dispatching.

In summary, based on existing studies, this paper provides an improvement on the construction of the uncertainty set of the traditional ARO. It avoids the drawback that an SO encounters difficulty in precisely obtaining the probability distribution of variables. The issue that the decision results based on RO are frequently overly conservative is also mitigated. In addition, historical data are employed to provide scientific data support for the construction of the uncertainty set of the ARO. Importantly, we leverage a meteorological clustering model and a GAN-based synthesized data-generation technique to improve the description accuracy of the uncertainty set.

Considering the strong correlation between electricity and carbon emissions, the MMGS low-carbon economic dispatching strategy will be further studied in terms of reductions resulting from carbon trading. In view of the need for a higher solution speed for the expansion of MMGSs, we will investigate more efficient solution methods in future work.

## Figures and Tables

**Figure 1 sensors-23-05350-f001:**
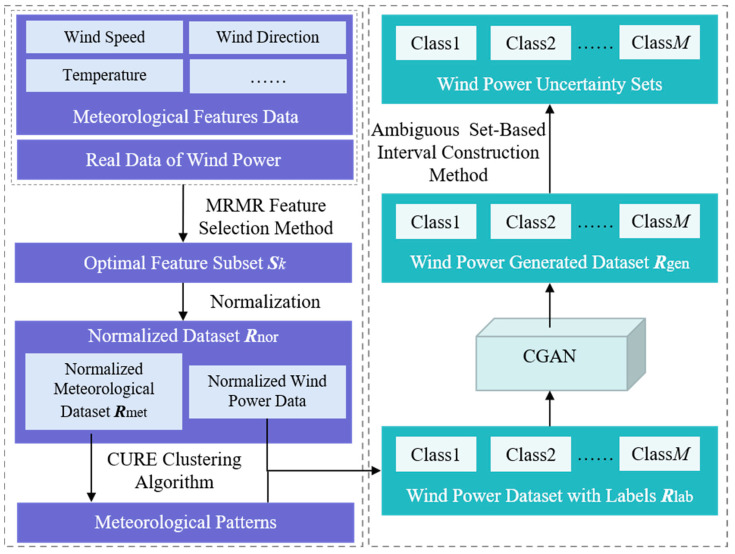
Principle framework of the proposed data-driven model.

**Figure 2 sensors-23-05350-f002:**
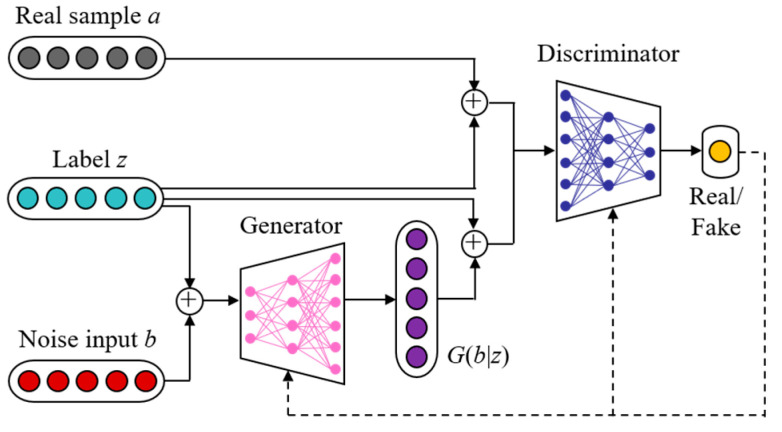
Basic structure of the CGAN model.

**Figure 3 sensors-23-05350-f003:**
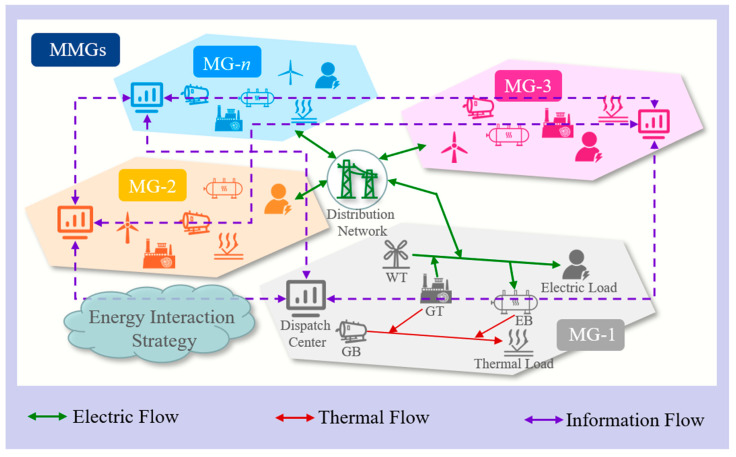
Structure diagram of the multi-microgrid system.

**Figure 4 sensors-23-05350-f004:**
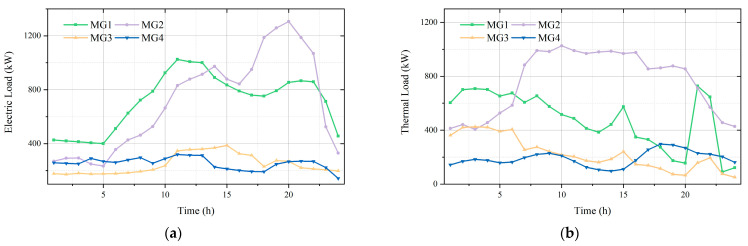
Load curves of each microgrid. (**a**) Electric load curves; (**b**) thermal load curves.

**Figure 5 sensors-23-05350-f005:**
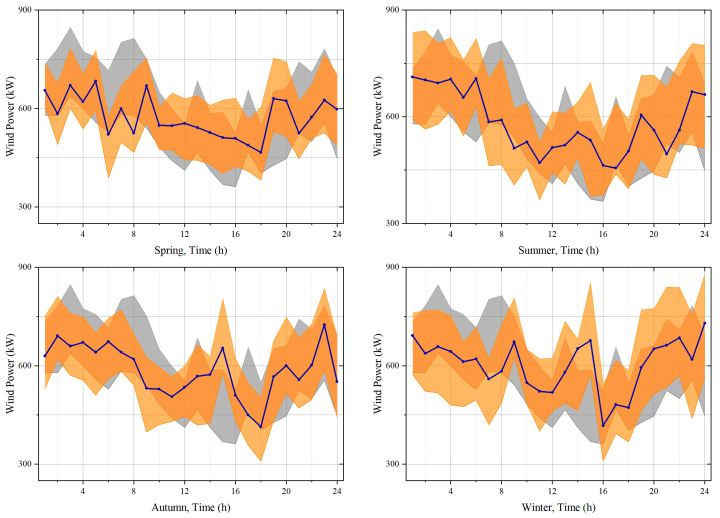
Comparison of the results of the wind-power interval estimation.

**Figure 6 sensors-23-05350-f006:**
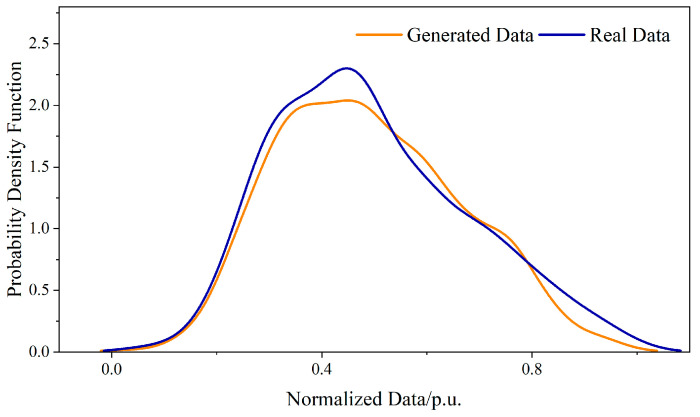
PDF of the real samples and the generated synthesized samples.

**Figure 7 sensors-23-05350-f007:**
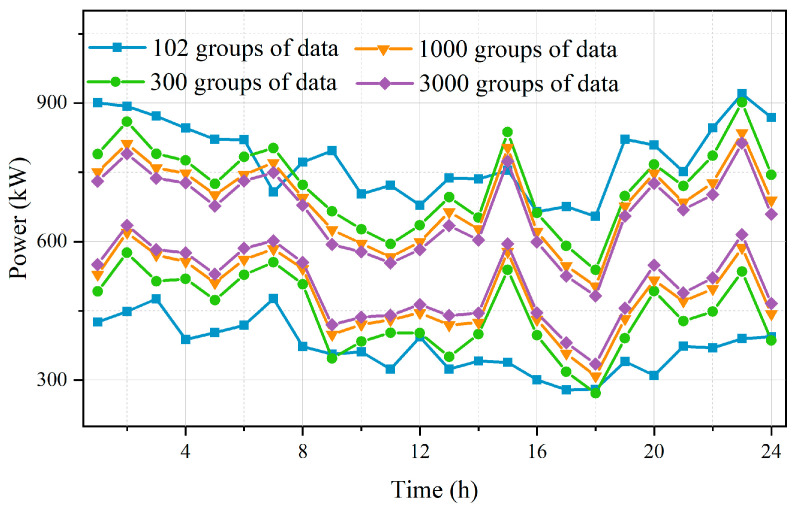
Comparison of uncertainty intervals for different sample sizes.

**Figure 8 sensors-23-05350-f008:**
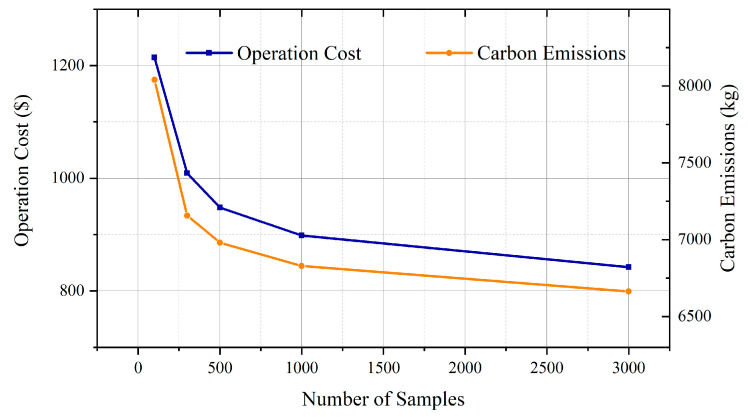
Relationship between sample quantity, operation costs, and carbon emissions.

**Figure 9 sensors-23-05350-f009:**
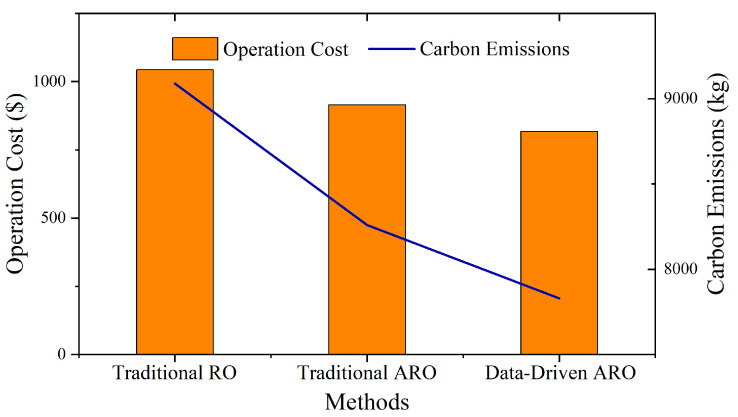
Comparison of dispatching results with different optimization methods.

**Figure 10 sensors-23-05350-f010:**
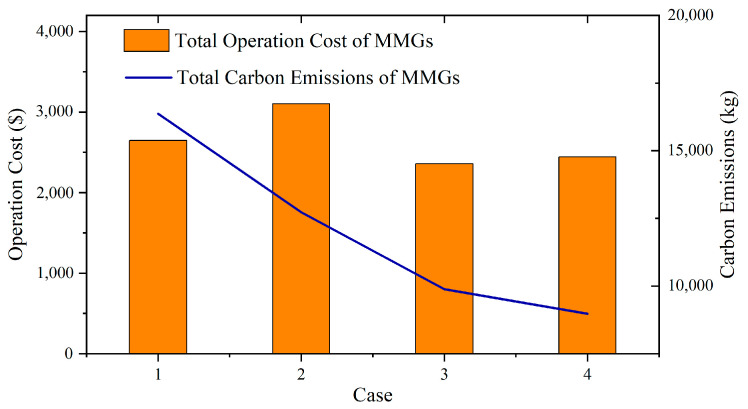
Comparison of four different dispatching cases of the MMGS.

**Figure 11 sensors-23-05350-f011:**
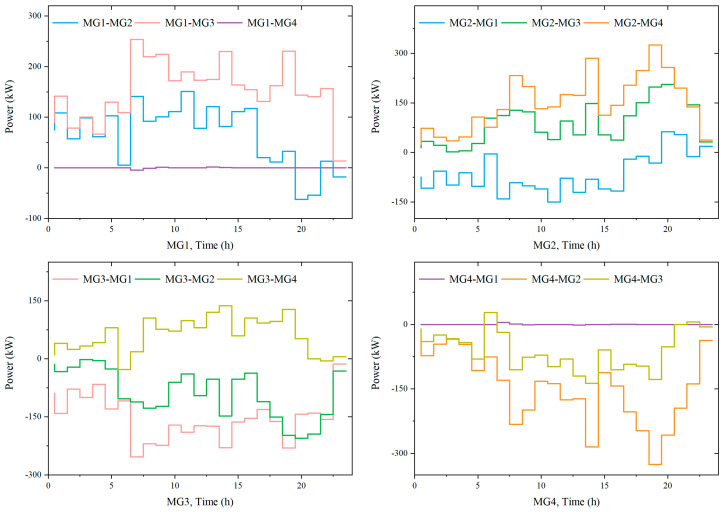
Optimization results of power interaction among MMGSs.

**Table 1 sensors-23-05350-t001:** Device parameters of microgrids.

Parameter	Value	Unit	Parameter	Value	Unit
$P_{\mathrm{Grid}}^{\max}$	800	kW	$P_{ij}^{\max}$	400	kW
$P_{\mathrm{GT}}^{\min}/P_{\mathrm{GT}}^{\max}$	50/400	kW	$P_{\mathrm{GB}}^{\min}/P_{\mathrm{GB}}^{\max}$	0/400	kW
$P_{\mathrm{EB}}^{\min}/P_{\mathrm{EB}}^{\max}$	0/400	kW	$R_{\mathrm{GT}}^{\min}/R_{\mathrm{GT}}^{\max}$	6.67/6.67	kW·h^−1^
$\eta_{\mathrm{GT}}$	0.35	-	$\eta_{\mathrm{WH}}$	0.83	-
$\eta_{\mathrm{HE}}$	0.8	-	$\eta_{\mathrm{GB}}$	0.9	-
$\eta_{\mathrm{EB}}$	0.99	-	$H_{\mathrm{ng}}$	9.7	kW/m^3^
$l_{C}$	200	kg	ρ	0.25	-

**Table 2 sensors-23-05350-t002:** Operation cost coefficient of microgrids.

Parameter	Value	Unit	Parameter	Value	Unit
$\tau_{{\mathrm{CO}}_{2}}$	0.03664	USD/kg	$\tau_{\mathrm{GT}}^{\mathrm{up}}/\tau_{\mathrm{GT}}^{\mathrm{dw}}$	0.03344	USD/kW
$\tau_{\mathrm{GB}}^{\mathrm{up}}/\tau_{\mathrm{GB}}^{\mathrm{dw}}$	0.01599	USD/kW	$\tau_{\mathrm{EB}}^{\mathrm{up}}/\tau_{\mathrm{EB}}^{\mathrm{dw}}$	0.01599	USD/kW
$\tau_{\mathrm{loss}}$	0.0756	USD/kW	$\tau_{\mathrm{gas}}$	0.5089	USD/m^3^

**Table 3 sensors-23-05350-t003:** Electricity price of microgrids.

Time Period	Day-Ahead Stage		Real-Time Stage	
	Purchased Price (USD/kW)	Sold Price (USD/kW)	Purchased Price (USD/kW)	Sold Price (USD/kW)
(12:00–14:00, 19:00–22:00)	0.1309	0.06543	0.2617	0.03271
(08:00–11:00, 15:00–18:00)	0.07996	0.03998	0.1599	0.02006
(23:00–07:00)	0.02617	0.01309	0.05234	0.006543

**Table 4 sensors-23-05350-t004:** Comparison of the calculation time of different optimization methods.

Optimization Method	Traditional RO	Traditional ARO	Data-Driven ARO
Time/s	4.33	121.36	108.94

**Table 5 sensors-23-05350-t005:** The settings of different dispatching cases.

Case	Energy Interaction among Microgrids	Carbon Trading
1	×	×
2	×	√
3	√	×
4	√	√

**Table 6 sensors-23-05350-t006:** Details of the comparison of four different dispatching cases of the four microgrids.

MG	Case	Energy-Interaction Costs with External Networks (USD)	Energy-Interaction Costs among Microgrids (USD)	Device Dispatching Costs (USD)	Carbon Trading Costs (USD)	Carbon Emissions (kg)
MG1	1	1041.9549	-	24.9475	-	6628.06
	2	1102.6087	-	30.7238	127.145	4866.11
	3	659.1139	257.5627	25.08995	-	3216.09
	4	674.09201	250.142	22.8233	7.3277	3080.96
MG2	1	1543.4791	-	28.6927	-	8069.76
	2	1568.3277	-	39.4603	253.6474	7222.47
	3	1155.2733	254.8687	28.6927	-	4498.05
	4	1175.5901	246.2674	39.8747	40.3065	4311.03
MG3	1	−11.2314	-	34.2466	-	464.71
	2	−11.7853	-	38.5676	−29.3412	334.06
	3	200.99	−220.276	30.4563	-	784.87
	4	200.6745	−219.978	30.1524	−12.9717	780.85
MG4	1	−40.3239	-	24.1406	-	1204.60
	2	−27.09633	-	39.8659	−27.9585	302.70
	3	233.3495	−292.03617	23.5328	-	1385.80
	4	227.2329	−277.4536	32.9904	−3.9066	959.20

## Data Availability

No new data were created or analyzed in this study. Data sharing is not applicable to this article.

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
