# Peer review of "A Low-Carbon and Economic Dispatch Strategy for a Multi-Microgrid Based on a Meteorological Classification to Handle the Uncertainty of Wind Power"

_sensors, 2023, doi:10.3390/s23115350_

Round 1

Reviewer 1 Report

In the manuscript, the authors present a meteorological clustering model and a GAN-based synthesized data generation technique to handle the uncertainty in wind power. Then, a data-driven ARO model is applied to optimize the operation of a multi-microgrid (MMG).

This paper focuses on an exciting topic. The author needs to improve the entire content of this research.

1.  The methodology section's structure is unclear, especially in Section 3. The presentation must be more straightforward so the reviewer can see the author's contribution to this research field. Please explain in detail the role of the adjustable robust parameter Γ in the ARO model (in which constraints?). And how do the authors reformulate or solve problem ? The author can explain these issues in Appendix.

2. and a lot of editing errors, etc

3. Using the ARO model to determine the optimal schedule of a power system is familiar. A combination of a meteorological clustering model and a GAN-based synthesized data generation technique is the main contribution of this article.

4. Introduction and Section 2 are presented quite well.

There are still some issues to be addressed/clarified by the authors, as follows:

5. In section 3, the authors should explain the adjustable robust parameter Γ (upper bound, lower bound of uncertain parameters?).

6. In Section 4, the authors solve the ARO problem by the C&CG algorithm. This algorithm is not new, but it is not easy to use without explaining it. So, the authors should present this algorithm in detail or at least show some references about this algorithm.

7. In Fig.9, the authors compare three optimization methods: traditional RO, ARO, and data-driven ARO, in optimal cost and carbon emissions. The author should compare the calculation time of the three methods above to see the advantage of the data-driven ARO method.

Minor editing of the English language required

Reviewer 2 Report

The paper presents a two-stage adjustable robust optimization model based on the meteorological clustering to improve the uncertainty set of wind power.  There is enough literature review, the methodology is clear and the results are well presented. 

Author Response

Thanks for your approval of this article.

Reviewer 3 Report

The article is well written, but the most important issue of this research may be the verification of its innovation. It is necessary for the authors of the article "A Low-Carbon and Economic Dispatching Strategy for Multi-Microgrid Based on Meteorological Classification Handling the Uncertainty of Wind Power" to make the following comments to enrich the article and answer some questions.

1-     The title is unclear, I suggest to change the title to "A low-carbon and economic dispatch strategy for multi-microgrids based on a meteorological classification to handle the uncertainty of wind power." change it.

2-     A long abstract has been written. Observe the technical writing components of the abstract (general introduction, research introduction, research materials and methods, results, research gap and research future). Also, there is a need to edit the English language throughout the article. For this purpose, I have reviewed and modified the first few lines of the abstract, you can continue it.: In a modern power system, reducing carbon emissions has become a significant goal in mitigating the impact of global warming. Therefore, renewable energy sources, particularly wind power generation, have been extensively implemented in the system. Despite the advantages of wind power, its uncertainty and randomness lead to critical security, stability, and economic issues in the power system.:

3-     Use 5 keywords.

4-     The final claim in the abstract "Case studies indicated that the presented model Explain has great performances..." with data and figures.

5-     Avoid reporting multiple references as (1 and 2) or (4 and 5). Use up-to-date and pertinent research, as recommended In the following: 10.1016/j.renene.2022.11.006; 10.1002/mma.8271 ; 10.1007/s40095-021-00462-5; 10.1007/s40095-022-00503-7; 10.14710/ijred.2022.43838; 10.3390/su14063566

6-     Figure 3 requires clarification. Please add the necessary markings to Figure 4. Also in Figure 8

7-     Please double-check the units of the variables in Table 3, as they may require modification.

8-     In the Conclusions section, write a paragraph comparing the current research with the studies reviewed in the introduction. Please submit a proposal for additional research.

The article is well written, but the most important issue of this research may be the verification of its innovation. It is necessary for the authors of the article "A Low-Carbon and Economic Dispatching Strategy for Multi-Microgrid Based on Meteorological Classification Handling the Uncertainty of Wind Power" to make the following comments to enrich the article and answer some questions.

1-     The title is unclear, I suggest to change the title to "A low-carbon and economic dispatch strategy for multi-microgrids based on a meteorological classification to handle the uncertainty of wind power." change it.

2-     A long abstract has been written. Observe the technical writing components of the abstract (general introduction, research introduction, research materials and methods, results, research gap and research future). Also, there is a need to edit the English language throughout the article. For this purpose, I have reviewed and modified the first few lines of the abstract, you can continue it.: In a modern power system, reducing carbon emissions has become a significant goal in mitigating the impact of global warming. Therefore, renewable energy sources, particularly wind power generation, have been extensively implemented in the system. Despite the advantages of wind power, its uncertainty and randomness lead to critical security, stability, and economic issues in the power system.:

3-     Use 5 keywords.

4-     The final claim in the abstract "Case studies indicated that the presented model Explain has great performances..." with data and figures.

5-     Avoid reporting multiple references as (1 and 2) or (4 and 5). Use up-to-date and pertinent research, as recommended In the following: 10.1016/j.renene.2022.11.006; 10.1002/mma.8271 ; 10.1007/s40095-021-00462-5; 10.1007/s40095-022-00503-7; 10.14710/ijred.2022.43838; 10.3390/su14063566

6-     Figure 3 requires clarification. Please add the necessary markings to Figure 4. Also in Figure 8

7-     Please double-check the units of the variables in Table 3, as they may require modification.

8-     In the Conclusions section, write a paragraph comparing the current research with the studies reviewed in the introduction. Please submit a proposal for additional research.

Round 2

Reviewer 1 Report

The authors improved the article. They also provided accurate responses to the questions and comments. So, I can suggest an article for publication.

Reviewer 3 Report

The authors of the article have provided accurate responses to the questions and comments. So, I can suggest an article for publication.